# Enhancing Textbooks with Visuals from the Web for Improved Learning

**Janvijay Singh**[*][G]      **Vilém Zouhar**[E]      **Mrinmaya Sachan**[E]

[G] School of Interactive Computing, Georgia Institute of Technology

[E] ETH Zürich, Department of Computer Science

iamjanvijay@gatech.edu      {vzouhar,msachan}@ethz.ch

## Abstract

Textbooks are one of the main mediums for delivering high-quality education to students. In particular, explanatory and illustrative visuals play a key role in retention, comprehension and general transfer of knowledge. However, many textbooks lack these interesting visuals to support student learning. In this paper, we investigate the effectiveness of vision-language models to automatically enhance textbooks with images from the web. We collect a dataset of e-textbooks in the math, science, social science and business domains. We then set up a text-image matching task that involves retrieving and appropriately assigning web images to textbooks, which we frame as a matching optimization problem. Through a crowd-sourced evaluation, we verify that (1) while the original textbook images are rated higher, automatically assigned ones are not far behind, and (2) the precise formulation of the optimization problem matters. We release the dataset of textbooks with an associated image bank to inspire further research in this intersectional area of computer vision and NLP for education.

## 1 Introduction

Students use textbooks as one of the primary mediums of learning. It is thus imperative that textbooks are designed to provide a rich learning and engaging environment. Visuals enhance learning through a number of means, including the ability to retain information, as well as its ability to promote comprehension and knowledge transfer (Carney and Levin, 2002; Dimopoulos et al., 2003; Katsioloudis, 2007; Hibbing and Rankin-Erickson, 2008; Panjwani et al., 2009; Mayer, 2019). Automatic approaches for retrieving and assigning images from the web to textbook chapters can therefore assist textbook designers in the creation of better textbooks. However, this is a very challenging task

---

[*] Work done during an internship at ETH Zürich

[0] Code & data: github.com/eth-nlped/textbook-enrichment

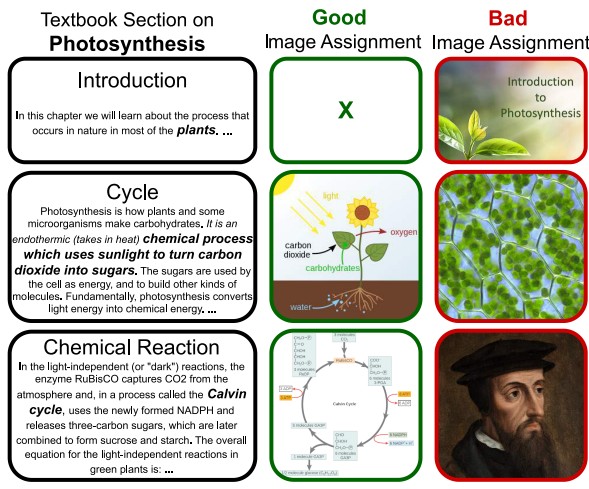

Figure 1: Illustration of *good and bad image assignments*. The first subsection does not require an image. The second subsection in the bad assignment has a related image but without a strong connection. Likewise, the last subsection has a picture of Calvin, which is related but does not have a high pedagogical value.

as an ideal illustrative visual should not only be related to the textbook material, but also have pedagogical value (see Figure 1).

Agrawal et al. (2010, 2011) already enhance textbooks with images from the Internet. They use similarity scores of textual captions from images and the text present in the textbooks for image assignment. This approach requires the image captions to be available. Beyond the problem of availability, the captions are also highly context dependent which reduces their utility in our setting. In this work, we aim to reinvigorate this area, and present and analyse a real dataset of textbooks. We do so by setting up the problem of image assignment using textbook and Wikipedia images.

In contrast to previous work, we rely on the recent advances in vision-language models, such as CLIP (Radford et al., 2021) and DALL-E (Ramesh et al., 2021). We analyze our dataset to gain insights into the organization of concepts and illustrative images within the textbooks. This analysis inspires the formulation of a new optimization

problem focused on modeling of illustrations in textbooks. The solution to this problem maximizes the coverage of the illustrations while minimizing redundancy during image-to-paragraph assigments. Our approach uses CLIP to retrieve and appropriately assign images to long-form text, particularly a textbook section. Then, the overall assignment is obtained in following stages:

1) Given a piece of text, produce a set of concepts that should be addressed by a visualization,
2) Given an image and a concept, determine their mutual relevance, and
3) Given a piece of text (e.g., a textbook section), produce an adequate assignment of images.

Each of the three sub-problems listed above can be solved with a variety of approaches — indeed we explore several variants, which we describe in Section 4. Because of the modularity of this approach, each of the sub-problems can be improved independently and adapted to the dataset in future work. Overall, we contribute the following:

- A dataset that contains text and images drawn from 35 textbooks covering math, business, social sciences, and science in addition to a secondary image bank of ∼312K images taken from Wikipedia (Section 3).
- Formalization of multiple textbook enrichment optimization goals (Section 4).
- Human evaluation and an in-depth examination of the possible failure modes and challenges of the proposed methods (Section 5).

## 2 Related Works

**Vision Language Models:** Alignment between texts and images has seen rapid progress recently with models such as CLIP (Radford et al., 2021) and DALL-E (Ramesh et al., 2021). However, their usages are limited to a short and specific text prompt to which the performance is usually quite sensitive. We focus on the problem of retrieving images for very long textual inputs, specifically a textbook section, where it is unclear which part of the text specifically describes the relevant image.

**Image Text Matching for Long Texts:** Recently, Wang et al. (2022a,b); Zeng et al. (2022) trained better language-vision representations with more nuanced associations, such as multiple vision tasks or finer image-text alignments. However, this progress is mainly confined to datasets, like MS-COCO (Lin et al., 2014) and Flickr-30K (Young

et al., 2014), that contain natural images and their captions, which are rather short. Additionally, Schneider et al. (2021) show that current multi-modal models perform poorly at retrieving relevant images for longer and more complex textual inputs. The reason for this poor performance is the pre-training on shorter and very specific image captions. This is a strong requirement to our work, which is focused on even longer text inputs. We explore the problem of assigning images to lengthy text, which highlights issues such as ensuring comprehensive coverage of concepts and avoiding redundant image illustrations. To move even closer to a real-world setting, we perform this task with actual textbooks and a human study.

**Enriching Text with Images:** The task of textbook enrichment was first explored by Agrawal et al. (2010, 2011), who assume that web images have associated relevant captions. We note that the caption of images are largely dependent on the context where the image was originally assigned.[1] The language of the caption may not even match the textbook's language. To alleviate all this, we do not assume that the images have associated relevant captions. Seo et al. (2015); Kembhavi et al. (2017); Lee et al. (2022) also studied associating images with textual information. However, their primary goal has largely been to comprehend the image content, and thus differs from our objective. Finally, there has been more past work on NLP applied to textbooks (Sachan and Xing, 2017; Sachan et al., 2017, 2020). However, the goal of these works also differ significantly from ours.

## 3 Dataset

We now present the process of curation and structure of our dataset with an analysis.

### 3.1 Data Collection

**OpenStax Books.** We source both the text and assigned images from 35 textbooks from an online textbook publisher openstax.org, covering four subjects: business, social sciences, sciences, and maths. Each textbook is organized into *chapters*, *sections*, *subsections*, and *paragraphs*. See their distribution in our dataset in Table 1. For each sub-

---

[1] For example, the same image of a poster from Wikipedia has different captions on different pages: *"The munitions industry heavily recruited women workers, as represented by the U.S. government's Rosie the Riveter propaganda campaign"* and *"J. Howard Miller's We Can Do It! poster from 1943s"*.

section of the textbook,[2] we identify the following key elements:

- **Text**: Raw text from the subsection.
- **Phrases**: Raw text from the subsection decomposed with overlapping sliding windows.[3]
- **Concepts**: Key concepts taught in the subsection are the *bolded* words or phrases, *headings*, *index* terms, and *key* terms (both marked explicitly in the book as such).
- **Images**: Image(s) assigned within subsection.

| Subject | Sections | Subsections | Images |
|---|---|---|---|
| Science | 1431 | 6775 | 5973 |
| Math | 461 | 4314 | 1177 |
| Social Science | 517 | 2463 | 1514 |
| Business | 553 | 2669 | 962 |
| All | 2962 | 16221 | 9626 |

Table 1: Summary of the number of sections, subsections, and assigned images in our dataset. We partition the chapters ($\sim$9.6K images) into *train* ($\sim$7.2K images), *dev* ($\sim$1.2K images), and *test* ($\sim$1.2K images) **splits**. While partitioning, we assign each book to a single split we ensure that all subjects are well represented in different splits and that there is minimal concept overlap.

**Wikipedia images.** To mimic the task of textbook enrichment, we use a dataset of images from Wikipedia that are relevant to the *concepts* in the OpenStax Books dataset. This dataset serves as a proxy for images from the web. We search for relevant Wikipedia articles for each concept, with a maximum of 20 articles retrieved per concept. From these articles, we extract images and their captions by searching the article in the WIT dataset (Srinivasan et al., 2021) or directly from the article. The final dataset has approximately $\sim$312K unique images included in the relevant articles.

**Image Bank.** We combine OpenStax Books images and the Wikipedia images to form the Image Bank. Our objective is to retrieve and assign relevant images from the Image Bank to each *section* that is present in the OpenStax Books dataset.

## 3.2 Dataset Analysis

We profile the dataset according to a series of questions, which will inform the problem formulation.

[2]Example of a subsection: openstax.org/books/concepts-biology/pages/5-1-overview-of-photosynthesis.
[3]A window-size of 75 tokens and an overlap-ratio of $1/3$.

**Q1. How are *concepts* distributed?** The patterns in concept mentions are similar across subjects (Figure 2b). The distribution of concepts within subsections (Figure 2a) reveals an average of 5.6 concepts per subsection. An average concept is mentioned 2.7 times within a subsection (Figure 2b) — that is, concepts are *infrequently* mentioned in the subsection. Notably, each section concept is mentioned in only 1.7 subsections on average (Figure 2c), emphasizing the high localization of concepts to specific subsections.

**Q2. What influences the number of assigned images in a subsection?** On average each section consists of 5.5 subsections and 3.3 assigned images (Table 1). To answer the question at hand, we conduct a regression analysis with the number of images in a subsection as the predicted variable, and the following as features:

- `concepts/words/paragraphs`: their total # in the subsection.
- `concepts_uniq`: # of unique concepts mentioned in the subsection.
- `%sec_concepts`: % of unique concepts from the section in the subsection.
- `%sec_concept`: % of total concept mentions from the section in the subsection.
- `%sec_words`: % of total words in the section which are in the subsection.
- `%sec_paragraphs`: % of paragraphs in the section in the subsection.
- `position` of the subsection in the section from 0 (beginning) to 1 (end).
- `subject` of the book.

Based on the results in Table 3, the number of assigned images to a subsection can be best predicted from total number of concepts, words, and paragraphs of the subsection. Unexpectedly, this is not true for the number of unique concepts. Furthermore, the position of the subsection within the section is negatively correlated to the image count — that is, the subsections located later in the section have fewer images. The subject of the book also impacts the subsection's image count, with differing coefficients for each subject. Overall, the regression model yields Pearson correlation of 0.59 with $p < 10^{-4}$ — a high degree of predictability.

**Q3. Are images exclusive to the assigned subsection?** We use CLIP's similarity scores for image-phrase relevance (detailed in Section 4). For each image in the textbook we distinguish between the

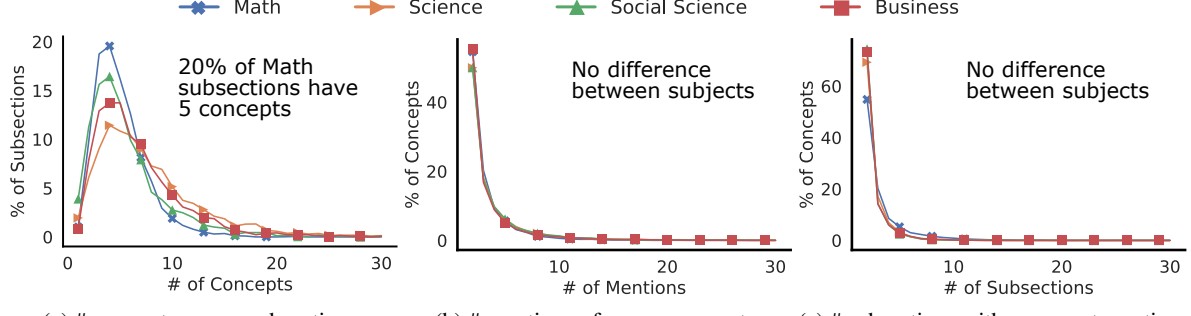

(a) # concepts across subsections.  (b) # mentions of across concepts.  (c) # subsections with a concept mention.

Figure 2: Distribution of concept-mentions. On average: (a) a subsection has 5.6 distinct concepts; (b) a concept is mentioned 2.7 times within a subsection; and (c) a concept from a section is mentioned in 1.7 of its subsections.

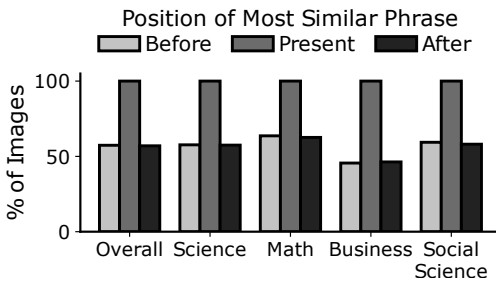

Figure 3: Proportion of images with the highest similarity scores to phrases from the subsection before, present, and after the gold subsection. Across all subjects, nearly 57% of the gold images have the highest CLIP similarity with at least one phrase the subsection before or after.

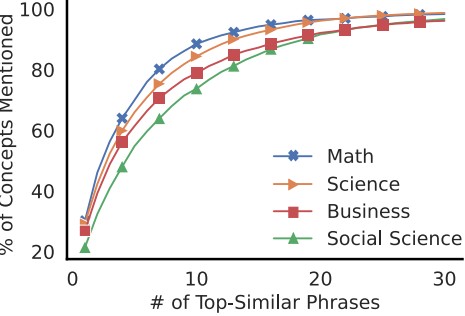

Figure 4: Proportion of concepts mentioned in top-$k$ most similar text-phrases to the gold-set of images in a subsection. On average for each subsection, 50% concepts are mentioned in the top-3 most similar text-phrases with the gold images.

present subsection and the one before and after. Then, we assign images to the subsection with the highest-matching phrase. The percentages of most-similar images in the before and after subsections is nearly equal (Figure 3), which is contrary to the intuition that the subsections after the current one would refer to the concepts in the image. The difference between the before/after and present subsections is the greatest in the business category, indicating uniqueness of images assigned to a particular subsection compared. Such uniqueness, as determined by the CLIP scores, is most absent in mathematics books. Overall, the images do not exclusively best-match the phrases from the gold-assigned subsection.

**Q4. Are concept mentions associated with assigned images?** We rank subsection phrases using CLIP similarity scores with subsection images. We use this ranking to calculate the percentage of concepts that were mentioned in the top-similar phrases associated with the gold images in the subsection. This way, we evaluate whether text-phrases with higher association to the gold-image also had more concept mentions. Indeed, there is a correspondence between the gold images and

phrase with concept mentions (Figure 4). This warrants further usage of CLIP scores as a measure for matching concepts to images.

## 4 Image Retrieval and Assignment

We first describe an image retrieval model and then formalize our task and the optimization approach. CLIP (Radford et al., 2021) is a state-of-the-art vision-language model trained on many image-caption pairs from the web by maximizing the dot-product similarity the image and caption encodings. We further fine-tune CLIP on image-text pairs from the OpenStax Books dataset.

### 4.1 Image Assignment Formulation

Our formulation focuses on the assignment of images to subsections.[4] We begin with notation:

$$\text{subsections } u \text{ in a section } s = \langle u_1, \ldots, u_{|s|} \rangle \quad (1)$$

$$\text{concepts } c \text{ in a subsection } u = \{c_1, \ldots, c_K\} \quad (2)$$

---

[4]We can assign images to the entire book by concatenating all chapters into one long "section". One can also assign images to paragraphs by treating them as a subsection each with a single paragraph.

We decompose the text of a subsection into *phrases* using a sliding window approach. These phrases may mention a particular *concept*:

$$\text{phrases } t \text{ in a subsection } u = \langle t_1, \ldots, t_L \rangle \quad (3)$$

Let $m(t_l, c_k)$ denote mention of concept $c$ in $t$

$$m(t_l, c_k) = \begin{cases} 1 & \text{if } c_k \text{ mentioned in } t_l \\ 0 & \text{otherwise} \end{cases} \quad (4)$$

For a fair comparison, we assign the same number of images to each subsection as in the gold assignment. This can also be automated with the image count prediction (Section 3.2/Q2).

## 4.2 Local Assignment

The most straightforward solution is to select an image for each subsection independently by maximizing the subsection text-image similarity. Specifically, we assign each subsection $u$, with an image $i \in \mathcal{I}$, which maximises the following function:

$$S(\{i\}, u) = \sum_{t \in u} \text{sim}(i, t) \quad (5)$$

Here, $\mathcal{I}$ denotes set of all the images in the image bank. Moreover, $\text{sim}(i, t)$ denotes probability (normalised dot-product similarity across images) of any image $i$ and certain phrase $t$ as given by the fine-tuned CLIP model.

While local assignment is fast and simple, our qualitative analysis reveals that it *lacks global coherence* and may assign images depicting overlapping concepts to the same section. For example, if every subsection mentions the concept "molecule", then all subsections can be assigned the same image of a molecule. This finding aligns with our previous results (Section 3.2/Q3) and is supported by the *redundancy* metrics in Section 5.

## 4.3 Global Assignment

The analysis in Section 3.2 revealed that the most-relevant phrases for gold images are not restricted to the assigned subsection and that the concepts are localized within their respective sections. Therefore, for better global coherence in our assignments, we assign images based on concepts rather than phrases. Specifically, we select a subset of images that covers most of the concepts (*coverage*) while avoiding overlaps (*redundancy*). To define *coverage* and *redundancy* functions for concepts in a section, we first define a boolean function for image, $i$, covering a concept, $c$, as follows:

$\text{cov}(c, i) = \mathbb{I}_{\text{sim}(i,t) \geq \tau} \cdot m(t, c)$. Informally, $\text{cov}(c, i)$ is 1 iff concept $c$ is covered by image $i$, otherwise 0. Next, we formalize the coverage and redundancy.

**Coverage.** Coverage for a section $s$ and a subset of images $\mathcal{I}'$ is the number of unique concepts in section $s$ which are covered by images in $\mathcal{I}'$:

$$C(\mathcal{I}', s) = \left| \{ c \in s \mid \exists i \in \mathcal{I}' : \text{cov}(c, i) = 1 \} \right| \quad (6)$$

**Redundancy.** Redundancy of a section $s$ and a a set of images $\mathcal{I}'$ is the total number of times concepts in $s$ are multiply covered:

$$R(\mathcal{I}', s) = \sum_{c \in s} \sum_{i \in \mathcal{I}'} \text{cov}(c, i) - C(\mathcal{I}', s) \quad (7)$$

We now introduce the concept of set submodularity which will be necessary for proving approximation bounds of the optimization.

**Definition 4.1.** *A function $f$ is said to be **set submodular** if and only if $\forall A \subseteq B \ \forall x : f(A \cup \{x\}) - f(A) \geq f(B \cup \{x\}) - f(B)$. Informally, the function yields diminishing returns for item $x$.*

**Theorem 4.2.** *The coverage function $C$ is set submodular. That is for $\mathcal{I}''$ such that $\mathcal{I}'' \subseteq \mathcal{I}' \subseteq \mathcal{I}$ and $i \in \mathcal{I}$ it holds that $C(\mathcal{I}'' \cup \{i\}) - C(\mathcal{I}'') \geq C(\mathcal{I}' \cup \{i\}) - C(\mathcal{I}')$.*

*Proof.* Let $\mathcal{C}'$ and $\mathcal{C}''$ be sets of concepts from a subsection covered by images in $\mathcal{I}'$ and $\mathcal{I}''$ respectively. As per the definition of $C$, we have $\mathcal{C}'' \subseteq \mathcal{C}'$. Let $\mathcal{C}_i$ be the concepts covered by image $i$.

$$
\begin{aligned}
C(\mathcal{I}' \cup \{i\}, s) - C(\mathcal{I}', s) = & \quad (8) \\
= \left| \mathcal{C}' \cup \mathcal{C}_q \right| - \left| \mathcal{C}' \right| & \quad \text{(from def.)} \quad (9) \\
\leq \left| \mathcal{C}'' \cup \mathcal{C}_q \right| - \left| \mathcal{C}'' \right| & \quad \text{(from } \mathcal{C}'' \subseteq \mathcal{C}') \quad (10) \\
= C(\mathcal{I}'' \cup \{i\}, s) - C(\mathcal{I}'', s) & \quad \text{(from def.)} \quad (11)
\end{aligned}
$$

$\square$

**Theorem 4.3.** *The negative redundancy function $-R$ is set submodular. That is for $\mathcal{I}''$ such that $\mathcal{I}'' \subseteq \mathcal{I}' \subseteq \mathcal{I}$ and $i \in \mathcal{I}$ it holds that $-R(\mathcal{I}'' \cup \{i\}) + R(\mathcal{I}'') \geq -R(\mathcal{I}' \cup \{i\}) + R(\mathcal{I}')$.*

*Proof.* Similarly to Theorem 4.2, for redundancy function $R$ we observe that:

$$
\begin{aligned}
R(\mathcal{I}' \cup \{i_q\}, s) - R(\mathcal{I}', s) = & \quad (12) \\
= \left| \mathcal{C}' \cap \mathcal{C}_q \right| & \quad \text{(from def.)} \quad (13) \\
\geq \left| \mathcal{C}'' \cap \mathcal{C}_q \right| & \quad \text{(from } \mathcal{C}'' \subseteq \mathcal{C}') \quad (14) \\
= R(\mathcal{I}'' \cup \{i_q\}, s) - R(\mathcal{I}'', s) & \quad \text{(from def.)} \quad (15)
\end{aligned}
$$

$\square$

**Observation 4.4.** *Both $C$ and $R$ are monotone.*

For the **global assignment**, we choose images $\mathcal{I}' \subseteq \mathcal{I}$ such that the following is maximised:

$$G(\mathcal{I}', s) = C(\mathcal{I}', s) - R(\mathcal{I}', s) \qquad (16)$$

$G$ (Equation 13) is a submodular function because it is a sum of two submodular functions. Finding the optimal solution to $G$ is NP-hard (Lovász, 1983). However, since $G$ is a submodular function, greedy algorithm can lead to fairly good $1 - 1/e \approx 63\%$-approximation of optimisation of $G$ under cardinality constraints $|\mathcal{I}'| \leq B$, where $B$ is the budget (Nemhauser et al., 1978). Once images $\mathcal{I}'$ are greedily computed for a section, we assign an image $i \in \mathcal{I}'$ to the subsection, $u$, which maximises $C(\{i\}, u)$; to the subsection in which the image $i$ covers the most concepts.

### 4.4 Joint Assignment

The local assignment captures relevance while the global one also captures redundancy. To optimize both of them, we formulate the following objective with a trade-off hyper-parameter $\beta \geq 0$:

$$\begin{aligned} J(\mathcal{I}', s) &= S(\mathcal{I}', s) + \beta \cdot G(\mathcal{I}', s) \qquad (17) \\ &= \sum_{i \in \mathcal{I}'} \sum_{t \in s} \mathsf{sim}(i, t) + \beta \cdot G(\mathcal{I}', s) \end{aligned}$$

Note that $J$ is a submodular function, since it is the sum of two other submodular functions: $\beta \cdot G$ and $S$. The former is submodular due to the non-negativity of $\beta$ and previously proven submodularity of $G$. The submodularity of $S$ is proven below.

**Theorem 4.5.** *The local assignment function $S$ is set submodular. That is, for a set of images $\mathcal{I}'$ and $\mathcal{I}''$ such that $\mathcal{I}'' \subseteq \mathcal{I}' \subseteq \mathcal{I}$ and any image $i \in \mathcal{I} - \mathcal{I}'$ it holds that $S(\mathcal{I}'' \cup \{i\}) - S(\mathcal{I}'') \geq S(\mathcal{I}' \cup \{i\}) - S(\mathcal{I}')$.*
*Proof.*

$$S(\mathcal{I}' \cup \{i\}, s) - S(\mathcal{I}', s) = \sum_{t \in s} \mathsf{sim}(i, t) \quad (18)$$

$$\leq S(\mathcal{I}'' \cup \{i\}, s) - S(\mathcal{I}'', s) \qquad (19)$$

$\square$

Considering the submodularity of $J$, we select images greedily, similarly to optimizing $G$. Once a set of images, $\mathcal{I}'$, is greedily computed for a section, we assign an image $i \in \mathcal{I}'$ to the subsection, $u$, which maximises $S(\mathcal{I}', u) + \beta \cdot C(\mathcal{I}', u)$; to the subsection in which the image $i$ covers the most

concepts and has most similar text. Note that the local and global assignments are specific cases of this formulation. This formulation achieves our desideratum — images are assigned to specific subsections also with global context consideration.

## 5 Human Evaluation

The goal of improving textbooks is to help students learn. Testing a wide range of models directly by monitoring learning progress would require a very expensive long-term evaluation. Instead we turn to an intrinsic crowd-sourced evaluation where we ask teachers what they think about the qualities of the assignment.

### 5.1 Setup

We selected 32 crowd-workers from Prolific who are native English speakers and work in education. We compare 4 different assignments: the gold one by a human and three automatic ones (Section 4). Each participant is assigned a single section and evaluates all 4 systems on this section. This methodology may cause an unwanted priming effect, which we address in Section 5.2. We chose this setup deliberately because a lot of the annotation time is spent on reading the section text and we wanted to share this cost by annotating multiple assignments at once. Our evaluation consists of close-ended (limited number of answers) questions that pertain to both the local (subsection) and global (section) textual context (full annotation guidelines are in Appendix A):

**Local evaluation:**
- Is this image relevant to educational concepts described in this subsection text?
- Is this image redundant compared to previous images in this subsection?
- What is the type of this image?

**Global evaluation:**
- Is this image relevant to educational concepts described in this section?
- Is this image redundant compared to previous images in this section?
- Is this image didactically useful for explaining this section text?

The annotators first answer the local questions for all images and then the global questions. This way we made sure that they scanned the entire section and had a some overview of all images and how they relate to each other before answering the

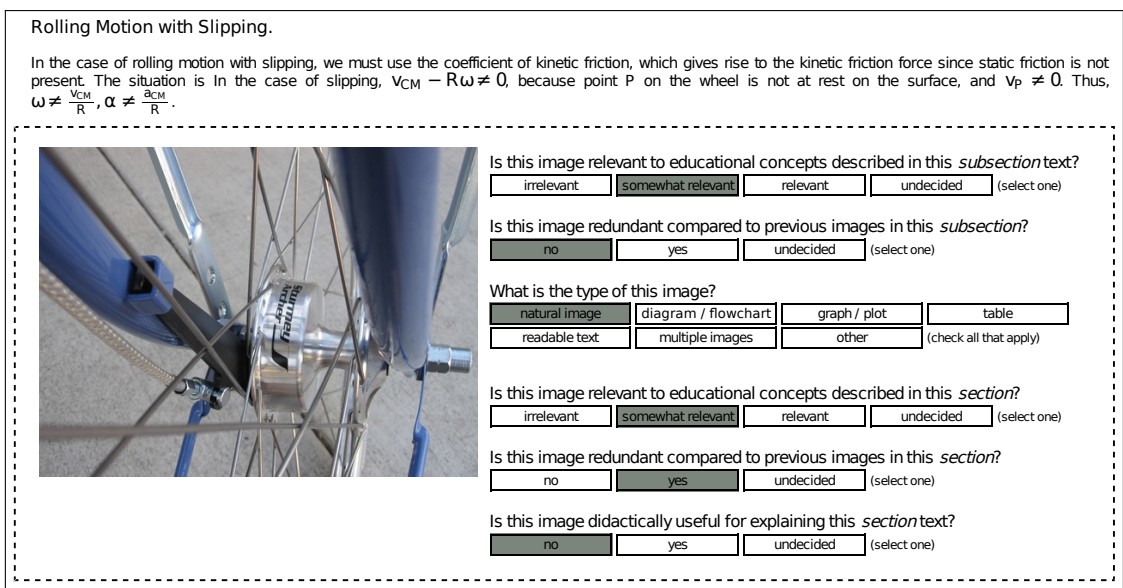

Figure 5: Annotation window for a single image within a subsection (there may be multiple images in a subsection). Note that, the image featured is not a suitable choice to illustrate this particular subsection.

global questions. The annotation pipeline (for 4 assignments, local/global) is shown in Figure 6 and the user-interface in Figure 5.

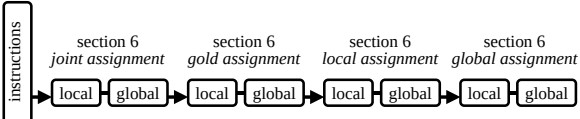

Figure 6: The annotator is presented with the same section but with randomized image assignment order.

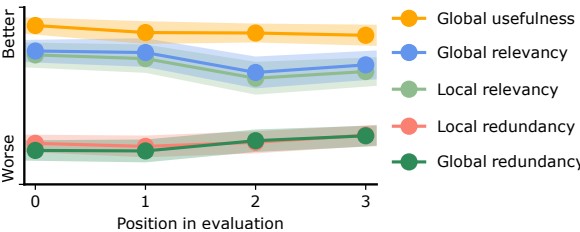

Figure 7: Average score for all categories as dependency of the position in evaluation. Color bands show 90% t-test confidence intervals.

## 5.2 Evaluation Results

We first verify the evaluation setup validity by checking whether the position in the evaluation queue has an effect on the evaluation scores. Recall that the annotators were shown the same textbook section with 4 alternate images assignments. While Figure 7 shows some variance along the independent variable of evaluation position, the differences are not significant, justifying our evaluation setup.

We then focus on two most important evaluation criteria: relevancy and redundancy. The results for those, shown in Figure 8, clearly show preference for the Gold image assignment, suggesting that automatic assignment is still inferior to humans. From the automatic methods, the Local performs the best relevancy-wise. However, it is outperformed by Joint with respect to redundancy. We also note that there is very little difference between Local and Global evaluation category. This may be caused by evaluation bias (i.e. annotators are likely to give similar score for both local and global questions).

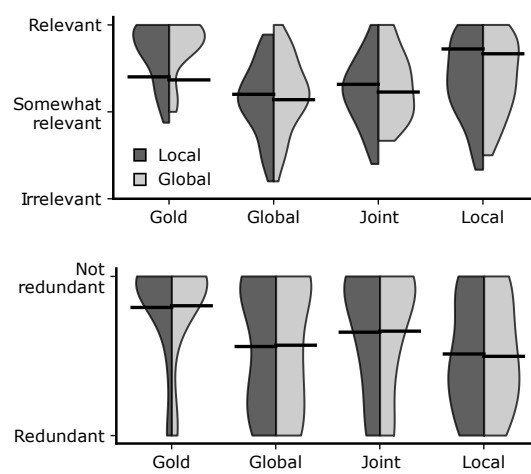

Figure 8: Local & global relevancy and redundancy across 4 image assignment modes. Gold is overall the best, though Local is better at relevancy.

Next, we examine all the remaining evaluation categories in Table 2. While Gold is the best across

all, the significance of the difference varies. The Joint assignment is never the worst, suggesting it to be a robust choice.

| Evaluation Category | Ordering |
|---|---|
| Local relevancy | Go $>_{0.001}$ L $>_{0.075}$ J $>_{0.163}$ Gl |
| Local redundancy | Go $>_{0.008}$ J $>_{0.288}$ Gl $>_{0.293}$ L |
| Global relevancy | Go $>_{0.001}$ L $>_{0.179}$ J $>_{0.106}$ Gl |
| Global redundancy | Go $>_{0.011}$ J $>_{0.337}$ Gl $>_{0.307}$ L |
| Global usefulness | Go $>_{0.001}$ J $>_{0.383}$ L $>_{0.211}$ Gl |

Table 2: Ordering of assignment modes with numbers showing one-sided Welsch's t-test p-value. Gold - Go, L - Local, J - Joint, Gl - Global.

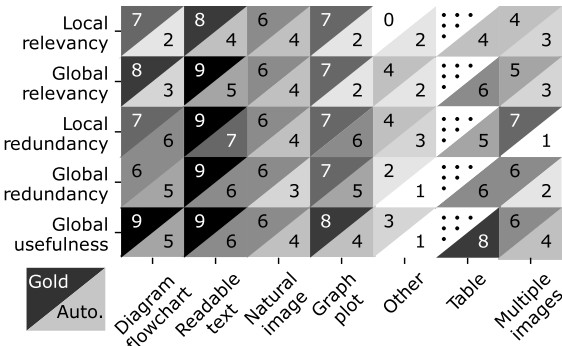

Figure 9: Category-wise gold and automatic image assignment scores (0-9); higher scores indicate better performance. In each cell, the top-left corner displays the score for the gold assignment, while the bottom-right displays the score for the automatic assignment (aggregated across all methods).

## 5.3 Qualitative Analysis

The Joint optimization method aims to reduce the reader's cognitive load compared to the Local method that aggregates scores from all text-phrases. The Local method results in repetitive covering of a single concept from the section with top-images. In contrast, Joint and Global assign images covering a wider range and variety of concepts in the section, enabling a greater level of text enrichment. Appendix B shows examples of assignments by these approaches. We remark that structured image types, such as graphs, multiple images, or those that are less identifiable, receive lower ratings systematically (Figure 9). We now elaborate on the two major limitations in our models.

**Varied domain of images.** One limitation of our approach is demonstrated in Figure 10 where it struggles to model non-natural images such as diagrams, graphs, and plots. These images often rep-resent abstract concepts, relationships, and events which cannot be well modeled by models like CLIP that are majorly trained on natural images.

**Long textual description of concepts.** Another source of error was that some concepts had long textual descriptions. For example, the description of *Stokes' theorem* spans multiple paragraphs. Learning to associate image with a part of text may lead to loose and spurious associations, resulting in poor downstream assignment performance. This highlights the need for vision-language representations that can effectively model long text descriptions and establish better image-text associations.

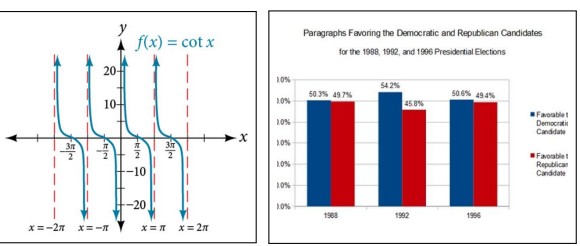

Figure 10: Our approach assigns left and right images to *Stokes' Theorem* and *Factors Affecting Engagement in Democracy*. In both cases, image is not specific enough and only very loosely relevant to the subsection text. This underscores the weakness of our model, which learns relevance with limited textual context.

## 6 Conclusion and Future Work

We presented a dataset and a new task of enriching textbooks with visuals from the web. We proposed several technical solutions for this problem using neural image retrievers combined with a new assignment optimization setup. Annotations by workers in the education industry verified that, even though the human assignment is still of the highest quality, the automatic assignments are not far behind. There are multiple venues for making further progress on this problem. First, individual concept importances and text-image relevance models could be improved and plugged into the existing algorithms. The varying domains of images and lengthy textual descriptions of concepts present challenges that could pave the way for exploring new approaches for learning image-text associations. Professional textbook designers can be included to further refine the assignment optimization objective and pose this as a human AI collaboration problem.

## Acknowledgments

We thank the anonymous reviewers for their feedback on our paper. MS acknowledges support from the Swiss National Science Foundation (Project No. 197155), a Responsible AI grant by the Haslerstiftung; and an ETH Grant (ETH-19 21-1) for this work.

## Limitations and Ethics Statement

While automatically enhancing textbooks with images holds promise, we point out:

- **Image selection bias**: Images from the web are at risk of being biased because they do not necessarily come from the same distribution as textbook graphics. However, images from Wikipedia are possibly more suitable for this purpose because they are of encyclopedic nature.
- **Intellectual property**: Practitioners who use our automatic image assignment method for textbooks should take care to always follow the associated copyrights and attributions.
- **Pedagogical usefulness**: While we employed workers to intrinsically judge the quality of the assignments, the results should be replicated with an extrinsic evaluation (beyond the scope of our study) which also considers the impact on student learning and information retention.
- **Quality control**: The target audience of textbooks are students, who are a sensitive group. In the current formulation, the optimization will always produce *some* assignment but there is no mechanism for quality assurance. This *could* result in inappropriate images being assigned and expert human scrutiny should be employed.

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

| | Coef. | SE | *T*-Val. | *p* |
|---|---|---|---|---|
| intercept | -0.279 | 0.020 | -13.753 | ⋆⋆ |
| concepts | -0.003 | 0.002 | -1.179 | |
| concept_mentions | 0.005 | 0.001 | 7.519 | ⋆⋆ |
| words | 0.001 | 0.000 | 27.405 | ⋆⋆ |
| paragraphs | 0.011 | 0.003 | 3.879 | ⋆⋆ |
| %sec_concepts | -0.001 | 0.001 | -1.065 | |
| %sec_c._mentions | 0.002 | 0.001 | 2.417 | ⋆ |
| %sec_words | 0.002 | 0.002 | 1.591 | |
| %sec_paragraphs | 0.002 | 0.002 | 1.438 | |
| position | -0.001 | 0.000 | -5.630 | ⋆⋆ |
| subject_math | 0.260 | 0.021 | 12.201 | ⋆⋆ |
| subject_science | 0.520 | 0.018 | 28.817 | ⋆⋆ |
| subject_social | 0.152 | 0.022 | 6.83 | ⋆⋆ |

Table 3: Regression analysis for predicting the number of images in a subsection. The following variables are significant predictors: concept_mentions, words, paragraphs, position, and subject of a subsection. **SE** is standard error, ⋆⋆ is $p < 10^{-5}$, and ⋆ is $p < 0.05$.

# A  Annotation Guidelines

The goal of your task is to help us evaluate our research on Artificial Intelligence (AI) based methods for creating more engaging and effective textbooks for students. You will be evaluating the performance of various AI-based methods that use images obtained from the internet to assign images to sections of a children's textbook.

In the first stage of your evaluation, you will be provided with text from a section of the textbook that has been divided into multiple subsections. Each subsection will be enclosed within a bounding box and will include images at the end. Your task is to evaluate the relevance, usefulness, and type of each image with respect to the text and images only within the same subsection.

After you finish answering questions in the first stage, new questions will appear with all the images (before and after shown in below and above image), these will be about second stage of evaluation. In the second stage of the evaluation, you will be reviewing the same subsections and images as before. Your task is to evaluate the relevance, redundancy and usefulness of each image in relation to the text and images within all the subsections in this section. Here, you will be asked to answer the new questions. Note: Please make sure to read the subsections that do not have any images during the first stage, as they will be useful for answering the questions in the second stage.

Choices concerning relevancy questions:
- **relevant**: some concept described in the subsection/section text is picturised in the image;
- **somewhat-relevant**: image is loosely on the same subject/topic domain as the subsection/section text, but is not exactly about the same concepts;
- **irrelevant**: not at all related to the subsection/section text.
- **undecided**: unable to make a decision, maybe because of my low understanding of the text or image.

Choices for redundancy questions:
- **yes**: this image illustrates exactly the same concept as one of the previously shown images in the section/subsection;
- **no**: this image illustrates atleast some unique concepts as compared to the previously shown images in the section/subsection;
- **undecided**: unable to make a decision, maybe because I do not understand the concepts illustrated by this image.

Choices for the didactic-usefulness questions:
- **yes**: image will be appropriate for teaching the section to a student - I would like it here (with an appropriate caption) if I was learning from this textbook;
- **no**: image is not appropriate for teaching the section text;
- **undecided**: unable to make the decision, maybe because of my less understanding of the text or image."

## B    Images Assigned with Joint

- *Section* "Engagement in a Democracy" from textbook "American Government 3e"
    - *Subsection* "Why Get Involved?" – Fig. 11
    - *Sub.* "Pathways to Engagement" – Fig. 13
    - *Subsection* "Factors of Engagement" – Fig. 14
- *Section* "Systems of Gas Exchange" taken from textbook "Biology 2e"
    - *Subsection* "Learning Objectives" – Fig. 16.
    - *Subsection* "Direct Diffusion" – Fig. 19.
    - *Subsection* "Skin and Gills" – Fig. 20.
    - *Subsection* "Tracheal Systems" – Fig. 23.
    - *Subsection* "Mammalian Systems" – Fig. 17.
    - *Sub.* "Lungs: Bronchi and Alveoli" – Fig. 24.
- *Section* "Stokes' Theorem" taken from textbook "Calculus Volume 3"
    - *Subsection* "Stokes' Theorem" – Fig. 18.
    - *Subsection* "Stokes' Theorem Proof" – Fig. 25.
    - *Subsection* "Interpretation of Curl" – Fig. 15.
- *Section* "Corporate Law and Corporate Responsibility" taken from textbook "Business Ethics"
    - *Subsection* "The Advantages of Corporate Status" – Fig. 12.
    - *Subsection* "Balancing the Many Responsibilities of a Corporation" – Fig. 21.
    - *Subsection* "The Two Sides of the Corporate Responsibility Debate" – Fig. 22.

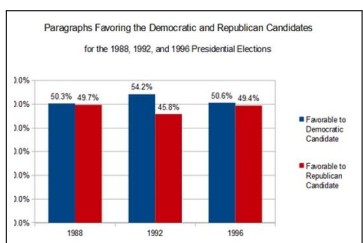

Figure 11: Image assignments for the "Why Get Involved?" subsection in "Engagement in a Democracy" section of the textbook "American Government 3e".

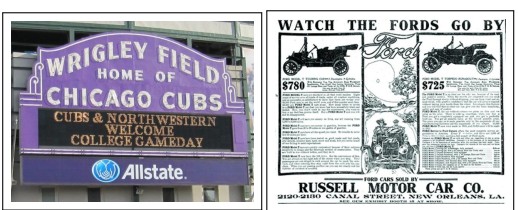

Figure 12: Image assignments for the "The Advantages of Corporate Status" subsection in the "Corporate Law and Corporate Responsibility" section of the textbook "Business Ethics".

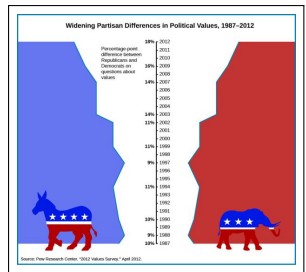
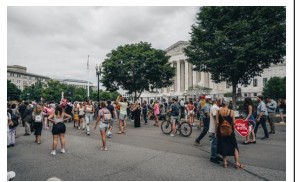
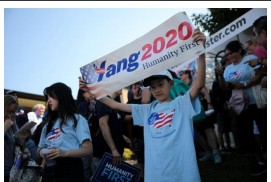

Figure 13: Image assignments for the "Pathways to Engagement" subsection in "Engagement in a Democracy" section of textbook "American Government 3e".

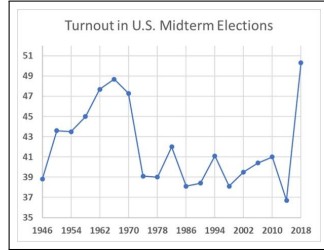

Figure 14: Image assignments for the "Factors of Engagement" subsection in "Engagement in a Democracy" section of the textbook "American Government 3e".

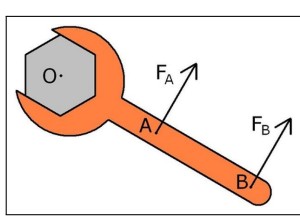
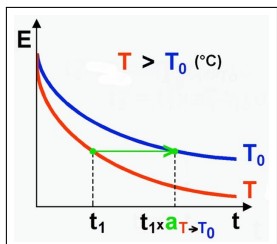

Figure 15: Image assignments for the "Interpretation of Curl" subsection in the "Stokes' Theorem" section of the textbook "Calculus Volume 3".

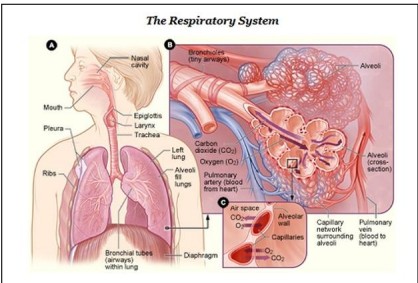

Figure 16: Image assignment for the "Learning Objectives" subsection in the "Systems of Gas Exchange" section of the textbook "Biology 2e".

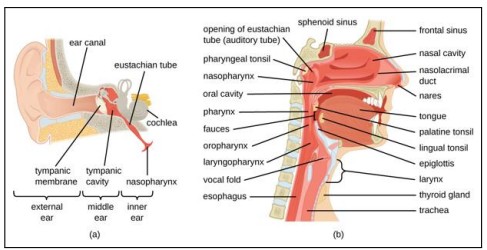

Figure 17: Image assignment for the "Mammalian System" subsection in the "Systems of Gas Exchange" section of the textbook "Biology 2e".

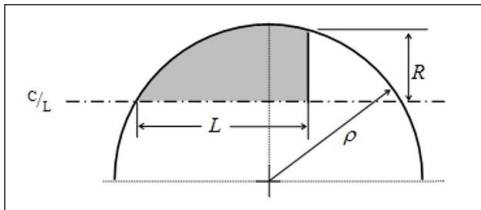

Figure 18: Image assignments for the "Stokes' Theorem" subsection in the "Stokes' Theorem" section of the textbook "Calculus Volume 3".

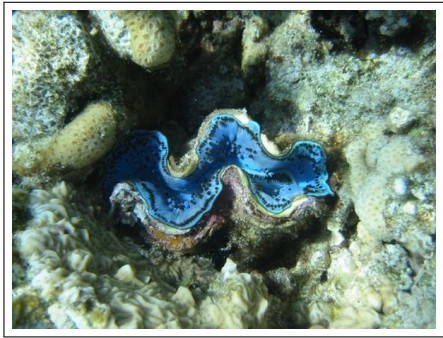

Figure 19: Image assignments for the "Direct Diffusion" subsection in the "Systems of Gas Exchange" section of the textbook "Biology 2e".

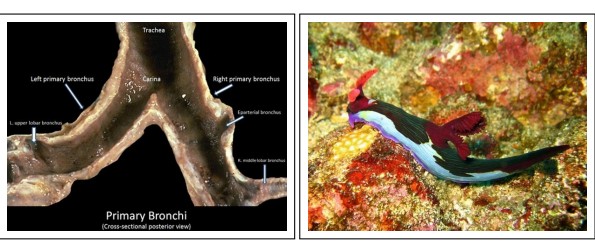

Figure 20: Image assignments for the "Skin and Gills" subsection in the "Systems of Gas Exchange" section of the textbook "Biology 2e".

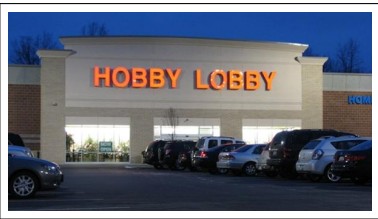

Figure 21: Image assignments for the "Balancing the Many Responsibilities of a Corporation" subsection in the "Corporate Law and Corporate Responsibility" section of the textbook "Business Ethics".

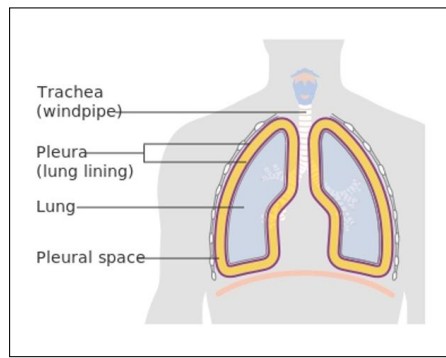

Figure 22: Image assignments for the "The Two Sides of the Corporate Responsibility Debate" subsection in the "Corporate Law and Corporate Responsibility" section of the textbook "Business Ethics".

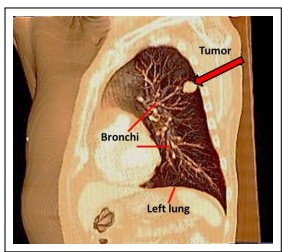

Figure 23: Image assignments for the "Tracheal Systems" subsection in the "Systems of Gas Exchange" section of the textbook "Biology 2e".

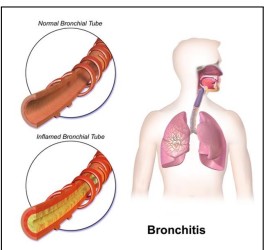

Figure 24: Image assignments for the "Lungs: Bronchi and Alveoli" subsection in the "Systems of Gas Exchange" section of the textbook "Biology 2e".

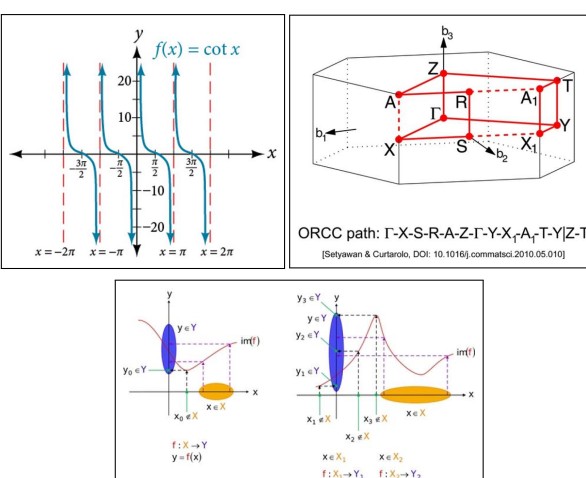

Figure 25: Image assignments for the "Stokes' Theorem Proof" subsection in the "Stokes' Theorem" section of the textbook "Calculus Volume 3".

## C   CLIP Fine-tuning

The CLIP model is composed of two parts: (1) the Image Encoder responsible for encoding the $p^{\text{th}}$ input image into a 512-dimensional vector $I_p$; and (2) the Text Encoder that encodes the $q^{\text{th}}$ input text into a 512-dimensional vector $T_q$. The model is trained with contrastive loss on a dataset of 400 million image-text pairs from the web. During the training, relevant image-caption pairs maximize $I_p \cdot T_q$ while $I_p \cdot T_q$ is minimize for unrelated $I_{p'}$ and $T_{q'}$. During our fine-tuning, we create mini-batches of image-text pairs extracted from the *subsections* in OpenStax Books Dataset.

During inference, we encode all images in the Image Bank as $\{I_1, I_2, \ldots, I_N\}$ and all text-queries belonging to a particular *subsection* as $\{T_1, T_2, \ldots, T_M\}$. Next, we get similarity scores for all images in the Image Bank with respect to the $k^{th}$ text-query by calculating $S^k = \langle I_1 \cdot T_k, I_2 \cdot T_k, \ldots, I_N \cdot T_k \rangle$. We then normalize $S^k$ to get $P^k = \text{SOFTMAX}(S^k)$. Finally, for each *subsection*, we compute the relevance scores of all images in the Image Bank by aggregating the $P^k$ values, resulting in $P = \text{AGG}(P^1, P^2, \ldots, P^M)$, where AGG is an aggregate function such as the mean of $N$-dimensional vectors.

In search for the best image retrieval model, we provide details for various methods of creating image-text pairs from a subsection.

### C.1   Evaluation Metrics

We now explain the techniques for the retrieval model evaluation. We consider the images initially assigned in the same subsection as gold-images for that particular subsection. Therefore, for each subsection, the Image Bank is categorized into *gold-images* and *not gold-images*. To gauge the retrieval quality of a subsection from a given retrieval approach, we use these metrics. Ultimately we use the average of all of them.

- **Recall@K**: fraction of gold-images retrieved in top-K retrievals.
- **Recall@R**: fraction of gold-images retrieved in top-R retrievals; where R is the number of gold-images in the subsection.
- **Precision@K**: fraction of retrieved images (K in total) that are gold-images.
- **Precision@R**: fraction of retrieved images (R in total) which are gold-images; where R is the number of gold-images in the subsection.
- **Mean Gold Rank**: average rank of each gold image given relevancy sorting.

### C.2   Zero-Shot CLIP

In the experiments discussed of this section, we use a pre-trained CLIP (without fine-tuning) to fetch relevant images for each subsection. The experiment results are presented in Table 4. For inference, we adopt a similar approach to the one described earlier. Below, we elaborate on the main differences in experimental settings for these studies:

1. **Concepts**: We use all the *concepts* from a subsection as its *text-queries* (each concept is a separate text-query) and use mean for AGG.
2. **Clustered-Concepts**: We use all the *concepts* from a subsection as its *text-queries* (each concept is a separate text-query). Next, we form clusters of text queries using $k$-means ($k = 10$) clustering on text encodings. For inference, we apply AGG on the relevance scores $P$ to text-queries belonging to each cluster. Using the cluster's aggregated relevance score, we retrieve one image at a time from each cluster in a round-robin fashion. This experiment tests if giving equal importance to various "concepts-clusters" leads to increased variation in retrieved results and better performance.
3. **Concatenated-Concepts**: We concatenate different *concepts* from a subsection together and use these concatenated phrases as the text-queries for each *section* and use mean as the *agg* function. This experiment tests if giving more context (multiple terms provide better con-

text about the *section* content) with input text-queries improves the performance.

4. **Phrases**: We follow the same setting as Exp C.2.1, except with overlapping phrases (each phrase is a separate text-query) drawn from sub-section's raw-text instead of the *concepts*.

## C.3 Fine-tuned CLIP

We now finetune CLIP with the experiment results are shown in Table 5.

1. **Concepts**: *Fine-tuning*: Let $\{i_1, i_2, \ldots, i_n\}$ and $\{t_1, t_2, \ldots, t_m\}$ respectively be the set of images and concepts included in a *sub-section* from OpenStax Books Dataset. We fine-tune CLIP using the following such 'correct' image-text pairs from all the sections in the *train split* of OpenStax Books Dataset: $\{(i_1, t_1), (i_1, t_2), \ldots, (i_1, t_m), (i_2, t_1), (i_2, t_2), \ldots (i_2, t_m), \ldots (i_n, t_1), \ldots, (i_n, t_m)\}$.
   *Inference*: We follow the exact same setting as Exp C.2.1 for inference; except that we compute scores across different data *splits*.

2. **Concatenated Concepts**: We follow the same setting as Exp C.3.1 except that, during both fine-tuning and inference, we concatenate concepts together to form text-queries for encoding (like in Exp C.2.3).

3. **Phrases**: We follow the same setting as Exp C.3.1, except that, both during fine-tuning and inference we use overlapping phrases drawn from subsection's raw-text instead of the *concepts*.

4. **Pre-fine-tune**: We pre-fine-tune the CLIP model on image-caption pairs from the Wikipedia images dataset. Following this, we perform fine-tuning and inference the same as that described in Exp C.3.2.

5. **Frozen Encoders**: We freeze the parameters of both text and image encoders except the last linear layers. Next, we carry out fine-tuning and inference same as Exp C.3.2.

6. **Frozen Encoders**: We freeze the parameters of both text and image encoders except the last linear and self-attention layers. Next, we carry out fine-tuning and inference same as Exp C.3.2.

Based on our experimentation, using results on the test split, we determine that the gold-mean rank can be improved through the following means: (a) fine-tuning CLIP, as it outperforms zero-shot; (b) concatenating concepts to provide more contextual

| Exp # | Precision | | | | | Recall | | | | | Gold Rank |
|---|---|---|---|---|---|---|---|---|---|---|---|
| | @1 | @5 | @20 | @100 | @R | @1 | @5 | @20 | @100 | @R | |
| C.2.1 | 4.38 | 3.02 | 1.57 | 0.64 | 3.21 | 1.54 | 5.24 | 9.68 | 18.53 | 3.21 | 22529 |
| C.2.2 | 2.07 | 1.88 | 1.22 | 0.54 | 1.96 | 0.75 | 3.35 | 7.83 | 15.76 | 1.96 | 22973 |
| C.2.3 | 6.34 | 3.36 | 1.79 | 0.67 | 3.76 | 2.23 | 6.52 | 12.48 | 20.99 | 3.76 | 19236 |
| C.2.4 | 3.93 | 2.52 | 1.35 | 0.60 | 2.99 | 1.50 | 4.42 | 8.66 | 17.47 | 2.99 | 17325 |

Table 4: Zero-shot CLIP results. Precision and recall scores (within the range of 0 to 1) are normalized to a scale of 0 to 100 for better readability. The worst possible gold rank is approximately 312K.

| Exp # | Split | Precision | | | | | Recall | | | | | Gold Rank |
|---|---|---|---|---|---|---|---|---|---|---|---|---|
| | | @1 | @5 | @20 | @100 | @R | @1 | @5 | @20 | @100 | @R | |
| C.2.1 | train | 4.35 | 2.58 | 1.50 | 0.65 | 3.08 | 1.58 | 4.38 | 9.27 | 18.89 | 3.08 | 22720 |
| | dev | 3.89 | 3.39 | 1.82 | 0.71 | 3.75 | 0.97 | 4.42 | 9.88 | 17.98 | 3.75 | 17981 |
| | test | 6.34 | 3.36 | 1.79 | 0.67 | 3.76 | 2.23 | 6.52 | 12.48 | 20.99 | 3.76 | 19236 |
| C.3.1 | train | 2.81 | 1.74 | 1.09 | 0.50 | 1.93 | 1.01 | 3.09 | 7.12 | 14.85 | 1.93 | 16526 |
| | dev | 3.89 | 2.40 | 1.27 | 0.48 | 2.47 | 0.86 | 2.94 | 7.43 | 13.31 | 2.47 | 13786 |
| | test | 3.03 | 2.04 | 1.05 | 0.50 | 2.57 | 1.00 | 3.52 | 6.75 | 16.21 | 2.57 | 14248 |
| C.3.2 | train | 7.05 | 4.50 | 2.41 | 1.02 | 5.15 | 2.61 | 7.77 | 15.15 | 29.12 | 5.15 | 5827 |
| | dev | 2.46 | 2.04 | 1.56 | 0.72 | 1.94 | 0.83 | 3.70 | 9.00 | 20.45 | 1.94 | 8806 |
| | test | 3.01 | 2.30 | 1.27 | 0.60 | 2.96 | 1.14 | 4.55 | 9.32 | 20.07 | 2.96 | 7478 |
| C.3.3 | train | 8.21 | 4.46 | 2.43 | 1.02 | 5.74 | 3.19 | 7.77 | 15.47 | 29.30 | 5.74 | 5169 |
| | dev | 5.26 | 3.72 | 2.14 | 0.92 | 3.82 | 1.46 | 5.49 | 12.26 | 25.47 | 3.82 | 6509 |
| | test | 4.96 | 3.97 | 2.00 | 0.85 | 3.80 | 1.69 | 7.41 | 13.79 | 26.02 | 3.80 | 6126 |
| C.3.4 | train | 3.96 | 2.57 | 1.60 | 0.79 | 3.11 | 1.53 | 4.55 | 10.14 | 22.78 | 3.11 | 6071 |
| | dev | 2.46 | 2.11 | 1.51 | 0.75 | 2.00 | 0.64 | 3.44 | 8.92 | 21.85 | 2.00 | 6208 |
| | test | 3.01 | 1.70 | 1.21 | 0.64 | 1.71 | 0.95 | 2.82 | 8.02 | 19.22 | 1.71 | 5995 |
| C.3.5 | train | 4.72 | 3.06 | 1.87 | 0.85 | 3.59 | 1.64 | 5.21 | 11.31 | 24.04 | 3.59 | 6315 |
| | dev | 4.56 | 2.25 | 1.58 | 0.72 | 3.23 | 1.79 | 3.57 | 9.89 | 20.12 | 3.23 | 8509 |
| | test | 3.29 | 1.97 | 1.26 | 0.63 | 2.65 | 1.35 | 3.40 | 7.95 | 18.84 | 2.65 | 8539 |
| C.3.6 | train | 4.29 | 2.68 | 1.64 | 0.78 | 3.01 | 1.65 | 4.90 | 10.43 | 22.88 | 3.01 | 7925 |
| | dev | 2.81 | 2.46 | 1.51 | 0.73 | 2.20 | 0.64 | 3.45 | 8.25 | 20.21 | 2.20 | 8900 |
| | test | 2.47 | 1.75 | 1.14 | 0.62 | 2.23 | 1.14 | 3.24 | 6.98 | 18.49 | 2.23 | 8383 |

Table 5: Results of experiment using Fine-tuned CLIP. Metrics shown here follow same format as in Table 4. The first multirow shows the zero-shot CLIP results from Exp C.2.1 and serves as a comparison baseline.

information, rather than using each concept as a separate text-query; (c) pre-finetuning on image-caption from the Wikipedia images; (d) fine-tuning all layers of CLIP, as opposed to only fine-tuning the last few layers. Ultimately, we select the model from Exp C.3.3 as our retrieval model, even though Exp C.3.5 achieved a slightly better gold-mean rank. This is because Exp C.3.5 relies on Wikipedia image caption pairs, which may not be readily available for all web images.