# OpenReview forum: "Enhancing Textbooks with Visuals from the Web for Improved Learning"
_EMNLP/2023/Conference — EMNLP 2023 Main_

### Official Review · Reviewer_r6d1 · 2023-08-04

**Soundness:** 5

**Excitement:**

4: Strong: This paper deepens the understanding of some phenomenon or lowers the barriers to an existing research direction.

**Paper Topic And Main Contributions:**

This paper studies an interesting topic of how to leverage visual images from the web to enrich the information of textbooks and help people develop a comprehensive understanding of them. The authors 1) curate a diverse dataset from textbooks and various web-source images, 2) conduct a thorough analysis of the constructed dataset, 3) sophisticated design an assignment procedure to assign collected images to the most relevant texts, 4) conduct comprehensive evaluations and analyses on the constructed datasets and assignment algorithms. Each step is of reasonable and constructive design supported by thorough analyses and statistics. The full paper presents a comprehensive and convincing study of "Enhancing Textbooks with Visuals from the Web".

**Reasons To Accept:**

1. The proposed dataset is carefully collected and curated, which benefits our community.

2. The analyses of the datasets are thorough, validating the usefulness of the datasets.

3. The image assignment procedure is well-designed, which can probably assign semantic-relevant images to texts.

4. Extensive evaluation also validates the contribution of the dataset and assignment algorithms.

5. If this dataset is open-sourced as the authors claim, the research areas, including educational AI and multimodal learning, could benefit.

**Reasons To Reject:**

I see no reason to reject this paper.

**Reproducibility:**

4: Could mostly reproduce the results, but there may be some variation because of sample variance or minor variations in their interpretation of the protocol or method.

**Reviewer Confidence:**

3: Pretty sure, but there's a chance I missed something. Although I have a good feel for this area in general, I did not carefully check the paper's details, e.g., the math, experimental design, or novelty.

---

> ### Author Rebuttal · Authors · 2023-08-27
>
> Thank you for appreciating the design of our study and for acknowledging potential benefits of our work in educational AI and multimodal learning!

---

### Official Review · Reviewer_Ymez · 2023-08-05

**Soundness:** 3

**Excitement:**

4: Strong: This paper deepens the understanding of some phenomenon or lowers the barriers to an existing research direction.

**Paper Topic And Main Contributions:**

The paper creates a database for textbook section to image selection. Authors used openstax as well as wikipedia images for creating the image dataset. Authors started with dataset insights and then proceeded to image assignment problem and error analysis.

**Questions For The Authors:**

How did the authors made sure about the quality of annotation ? It will make the paper stronger.


**Reasons To Accept:**

Authors provided a detailed explanation of their datasets through some good insights about concepts and image distribution

They also tried to understand if image assignment has a correlation with different features (length of subsection, concepts, etc)

The output shows that the results are interesting and the error analysis provides insights on where the improvement can be made for future.


**Reasons To Reject:**

I did not find any concrete reasons to reject the paper

**Reproducibility:**

4: Could mostly reproduce the results, but there may be some variation because of sample variance or minor variations in their interpretation of the protocol or method.

**Reviewer Confidence:**

3: Pretty sure, but there's a chance I missed something. Although I have a good feel for this area in general, I did not carefully check the paper's details, e.g., the math, experimental design, or novelty.

---

> ### Author Rebuttal · Authors · 2023-08-27
>
> Thank you for highlighting the contributions of this work and for your constructive feedback.
>
> **Q1**: We measured the micro-averaged inter-annotator agreement between 32 annotators across all quantitative answers (12,900 pairs) in two ways: (a) partial match - answers within 1 point of the annotation scale; and (b) exact answer match. Further, we annotated a portion of the assignments ourselves, covering five sections, each on a different subject. The annotators' agreement (82% for partial and 52% for exact) closely matched the agreement from our annotations (81% for partial and 57% for exact), suggesting the annotators' annotations were of good quality. We will include these details in the updated version.

---

### Official Review · Reviewer_XsLy · 2023-08-05

**Typos Grammar Style And Presentation Improvements:** N.A.
**Soundness:** 3

**Excitement:**

4: Strong: This paper deepens the understanding of some phenomenon or lowers the barriers to an existing research direction.

**Justification For Ethical Concerns:**

N.A.

**Missing References:**

N.A.

**Paper Topic And Main Contributions:**

This paper formalizes the problem of enhancing educational text with images from the web. The key contributions of the paper are as follows:

1: Formalizes the task beyond simply calling it similarity matching. Gives it a local and global perspective.
2: Creates a dataset of various educational topics.
3: Examines points of failures and areas of improvement in this work.

**Questions For The Authors:**

Did you consider doing comparison between using finetuned clip vs pretrained clip?

Also, please elaborate sliding window approach in line 303 e.g. size of window etc.

**Reasons To Accept:**

1: The paper is well written, well organized, easy to follow (except the math in lines 365 to 390 which was very dry but it may appeal to other audience).

2: The neatly curated dataset containing texts, phrases, intelligently derived key concepts and images.

3: Invaluable Indepth analysis of data asking key questions that help design a working solution for anyone that uses this data.

4: A fair assessment of their performance. I was not expecting their approach to match or beat human performance, but the performance of their 'joint' optimization was not too far from it either. I can see this work be easily extended in the setup where "Accept images for sections where automated setup is highly confident, and use human to rate or identify images for subsections where the automated setup fails or is not as confident" . This would still save valuable human time by (semi)automating the process.

Overall, this paper is a good introduction to the task, shares a useful dataset with its own analysis that will help other researchers jumpstart work in this area. I recommend we publish this.

**Reasons To Reject:**

N.A.

**Reproducibility:**

4: Could mostly reproduce the results, but there may be some variation because of sample variance or minor variations in their interpretation of the protocol or method.

**Reviewer Confidence:**

3: Pretty sure, but there's a chance I missed something. Although I have a good feel for this area in general, I did not carefully check the paper's details, e.g., the math, experimental design, or novelty.

---

> ### Author Rebuttal · Authors · 2023-08-27
>
> Thank you for your review and for acknowledging that the contributions of this paper will help researchers jumpstart work in this area.
>
> **Q1**: Yes, we provide a comparison between pre-trained and fine-tuned CLIP in sections 8.2 and 8.3 of the appendix. The results clearly suggested that the mean gold-rank (retrieval rank of gold-images for a subsection) with the fine-tuned CLIP is consistently better, even across various ways in which the input text is formulated.
>
> **Q2**: CLIP is pre-trained on image-caption pairs with a text limit of 77 tokens. We split the subsections' text into overlapping windows of 75 tokens, as mentioned in Footnote 3. This approach maintains the text granularity consistent with CLIP's pre-training. Moreover, overlapping windows ensure concepts are presented in diverse contexts. For instance, _"photosynthesis involves plants absorbing sunlight"_ offers more visual cues than _"this process is called photosynthesis"_. Because of this, instead of splitting the original text ABCDEFHI into non-overlapping chunks (ABC, DEF, GHI) we use overlapping chunks (ABC, BCD, CDE, …). We will highlight these details in the updated version.

---

### Meta-Review · Area_Chair_w9X1 · 2023-09-10

**Recommendation:** 5

**Metareview:**

In general, all reviewers agreed that this research is both sound and exciting.

**Summary of Reviewer Feedback and Discussion:**
- **Reviewer XsLy** appreciated the neatly-curated dataset and in-depth analysis thereof, as well as the paper's fair performance assessment.  They were curious about a few finer details in the paper, which the authors provided in the rebuttal and promised to include in the revised manuscript.  The reviewer then thanked them for providing these details.
- **Reviewer Ymez** liked the detailed explanation of the dataset and the authors' careful investigation of the underlying task.  They also appreciated the thorough evaluation, and were curious about a few details pertaining to annotation quality.  The authors provided these details in the rebuttal, and promised to include them in the updated manuscript.
- **Reviewer r6d1** liked the carefully-curated dataset and thorough analysis thereof.  They also thought that the authors' methods were well-designed, and that the authors had extensively evaluated their work.  They felt that if the data is made publicly available, it may benefit numerous research subfields.  In the rebuttal, the authors thanked them for their appreciation of the study and acknowledgement of these potential benefits.

---

### Decision · Program_Chairs · 2023-10-07

**Decision:**

Accept-Main

**Comment:**

In general, all reviewers agreed that this research is both sound and exciting.

**Summary of Reviewer Feedback and Discussion:**
- **Reviewer XsLy** appreciated the neatly-curated dataset and in-depth analysis thereof, as well as the paper's fair performance assessment.  They were curious about a few finer details in the paper, which the authors provided in the rebuttal and promised to include in the revised manuscript.  The reviewer then thanked them for providing these details.
- **Reviewer Ymez** liked the detailed explanation of the dataset and the authors' careful investigation of the underlying task.  They also appreciated the thorough evaluation, and were curious about a few details pertaining to annotation quality.  The authors provided these details in the rebuttal, and promised to include them in the updated manuscript.
- **Reviewer r6d1** liked the carefully-curated dataset and thorough analysis thereof.  They also thought that the authors' methods were well-designed, and that the authors had extensively evaluated their work.  They felt that if the data is made publicly available, it may benefit numerous research subfields.  In the rebuttal, the authors thanked them for their appreciation of the study and acknowledgement of these potential benefits.